

# Genome-wide identification and characterization of the *Hsp70* gene family in allopolyploid rapeseed (*Brassica napus* L.) compared with its diploid progenitors

Ziwei Liang, Mengdi Li, Zhengyi Liu and Jianbo Wang

State Key Laboratory of Hybrid Rice, College of Life Sciences, Wuhan University, Wuhan, China

## ABSTRACT

Heat shock protein 70 (Hsp70) plays an essential role in plant growth and development, as well as stress response. Rapeseed *(Brassica napus* L.) originated from recently interspecific hybridization between *Brassica rapa* and *Brassica oleracea*. In this study, a total of 47 *Hsp70* genes were identified in *B. napus* ($A_nA_nC_nC_n$ genome), including 22 genes from $A_n$ subgenome and 25 genes from $C_n$ subgenome. Meanwhile, 29 and 20 *Hsp70* genes were explored in *B. rapa* ($A_rA_r$ genome) and *B. oleracea* ($C_oC_o$ genome), respectively. Based on phylogenetic analysis, 114 Hsp70 proteins derived from *B. napus*, *B. rapa*, *B. oleracea* and *Arabidopsis thaliana*, were divided into 6 subfamilies containing 16 $A_r$-$A_n$ and 11 $C_o$-$C_n$ reliable orthologous pairs. The homology and synteny analysis indicated whole genome triplication and segmental duplication may be the major contributor for the expansion of *Hsp70* gene family. Intron gain of *BnHsp70* genes and domain loss of BnHsp70 proteins also were found in *B. napus*, associating with intron evolution and module evolution of proteins after allopolyploidization. In addition, transcriptional profiles analyses indicated that expression patterns of most *BnHsp70* genes were tissue-specific. Moreover, *Hsp70* orthologs exhibited different expression patterns in the same tissue and $C_n$ subgenome biased expression was observed in leaf. These findings contribute to exploration of the evolutionary adaptation of polyploidy and will facilitate further application of *BnHsp70* gene functions.

## INTRODUCTION

Taken as a whole, polyploidization has long been seen as a key force in the evolution of eukaryotic nuclear genomes, and about 70% of angiosperms have experienced relatively recent genome doubling in the form of polyploidy (*Masterson, 1994*). Polyploidy often shows morphological innovation, can provide the basic material for the origin of plant adaptation, and thus have a significant impact on plant species diversity (*Adams & Wendel, 2005*). As the most common type of polyploidy, allopolyploidy generated from hybridization of two formerly differentiated genomes usually from different species. The whole process of allopolyploidization event involves a series of molecular and physiological

Corresponding author
Jianbo Wang, jbwang@whu.edu.cn

adjustments. The onset of genomic shock occurred accompanied by the merger of two distinct genomes reunited in a common nucleus (*McClintock, 1984*). This collision among the subgenomes sometimes leads to subgenome bias and even to the dominance of one of a subgenome, thus affecting homologous exchanges, epigenetic regulation and gene expression (*Bird et al., 2018*). Meanwhile, some duplicate gene pairs (homologs) with similar or redundant functions are retained nonrandomly. Recent insights into subgenome bias and duplicate gene retention in polyploids contribute to sharpen researches of polyploid adaptation and provide great opportunities for trait improvement of polyploid species in agriculture (*Samans, Chalhoub & Snowdon, 2017*; *Bird et al., 2018*).

*B. napus* ($2n = 4x = 38$), an allotetraploid species, arose from gene duplication after natural hybridization between the diploid ancestors of *B. rapa* ($2n = 2x = 20$) and *B. oleracea* ($2n = 3x = 18$), followed by spontaneous chromosome doubling (*Chalhoub et al., 2014*). Compared to Arabidopsis, the genomes of all *Brassica* species have experienced a lineage-specific whole genome triplication (WGT) event, and rediploidization would follow that involved substantial genomic shock including gene loss and exchanges between genomes. With beneficial heterosis effect, *B. napus* has better adaptability to natural environment and can produce desirable traits in the agricultural environment. To date, *B. napus* is the third largest oilseed crops all over the world, with wide planting area and large yield. It is believed that polyploid lineages may have complex relationships with their diploid ancestors. *B. rapa* with 530 Megabase (Mb), *B. oleracea* with 630 Mb and *B. napus* with 849.7 Mb genomes have been released recently, which often used to elucidate genome evolution in angiosperms (*Chalhoub et al., 2014*). Also, the *Hsp70* gene family is well conserved in the evolution of angiosperms. Accordingly, it provides new chance to understand the origin and evolution of the *Hsp70* gene family in *Brassica* genomes.

Hsp70s, approximately 70kiloDalton (kDa) in size, are the most conserved and ubiquitous in heat shock proteins (HSPs) which are of great significance responsive to heat stress reaction (HSR) of plants (*Lindquist, 1986*; *Feder & Hofmann, 1999*). They function as molecular chaperones to prevent protein aggregation, deformation and promote protein refolding to repair damaged protein (*Wang et al., 2004*; *Mayer & Bukau, 2005*). Structurally, all Hsp70s have two major functional domains: highly conserved nucleotide-binding domain (NBD) and substrate-binding domain (SBD) that covered variable C-terminal 'lid' (*Lindquist, 1986*; *Zhu et al., 1996*). Despite the acidic SBD β insertion and longer C-terminal extension in Hsp110s, they share the same domain composition as classical Hsp70 and are therefore considered to be component of the Hsp70 family (*Liu & Hendrickson, 2007*). The *Hsp70* gene family has been widely reported in many plants, e.g., *A. thaliana* (18 genes); rice (32 genes); soybean (61 genes) and pepper (21 genes) (*Lin et al., 2001*; *Sarkar, Kundnani & Grover, 2013*; *Zhang et al., 2015*; *Guo et al., 2016*). Hsp70s have been confirmed to be indispensable in plant development, as well as associate with plant stress resistance. *AtHsp70-15*-deficient led to Arabidopsis plant dwarfing, leaf malformation and growth retardation (*Jungkunz et al., 2011*). Double-knockout mutations in *cpHsc70-1* (*At4g24280*) and *cpHsc70-2* (*At5g49910*) were defective to both female and male gametes (*Su & Li, 2008*). In resistance to abiotic stresses, cytosolic/nuclear Hsp70s in *A. thaliana* had both specific and redundant functions (*Leng et al., 2017*). Expression of *Hsp70* was

strongly correlated with thermotolerance in rice and can be considered potential biomarker in future rice breeding programs (*Ali et al., 2017*). Until now, little is known about the *Hsp70* gene family in *Brassica* species.

In this study, detailed studies of the *Hsp70* gene family of *B. napus* and diploid progenitors were carried out. All of the putative *Hsp70* orthologous gene members in the genomes of *B. napus* and its diploid progenitors were firmly identified using sequence similarity and Hsp70 specific domain. A comparative phylogenetic analysis was performed to infer the evolutionary relationships of the Hsp70 homologs of *B. napus* and its relatives, including *A. thaliana*, *B. rapa* and *B. oleracea*. Synteny and duplicated gene analysis among *B. rapa*, *B. oleracea* and *B. napus* genomes were investigated for better understanding the expansion patterns and evolution forces of the *Hsp70* gene family. We also explored *Hsp70* gene expression patterns in four tissues (stem, leaf, flower and silique). These thorough analyses of the *Hsp70* gene family in allopolyploid *B. napus* species and two diploid ancestors will help to better understand the molecular events after polyploidization, and will also open up more possibilities for further studies of *B. napus* and other polyploid species.

## MATERIALS & METHODS

### Identification of *Hsp70* gene members

In order to identify *Hsp70* gene members in the *B. napus* (cv. Darmor-*bzh*), *B. rapa* (cv. chiifu-401-42) and *B. oleracea* (var. *capitata* line 02–12), all proteins of *B. napus*, *B. rapa* and *B. oleracea* in the *Brassica* database (BRAD: http://Brassicadb.org/brad/) (*Cheng et al., 2011*) were performed Protein Basic Local Alignment Search Tool (BLASTp) algorithms using 18 Hsp70 protein sequences of Arabidopsis downloaded from the Arabidopsis Information Resource (TAIR10: https://www.Arabidopsis.org/) (*Lamesch et al., 2012*). The maximum $E$-value was >1e−5. Meanwhile, the Hidden Markov Model (HMM) profile of Hsp70 seed file (PF00012) was obtained from the Pfam database (https://pfam.xfam.org/) (*Finn et al., 2016*) and then submitted to search in HMMER (http://hmmer.org/) (*Eddy, 2009*) software locally. Proteins with Hsp70 domain were extracted from BRAD. Integrating the results of two methods, all redundant sequences were removed manually. The candidate sequences were further confirmed by the following databases: NCBI Conserved Domain Search database (CDD: http://www.ncbi.nlm.nih.gov/Structure/cdd/wrpsb.cgi/) (*Marchler-Bauer et al., 2015*), Simple Module Architecture Research Tool (SMART) database (http://smart.embl-heidelberg.de/) (*Letunic et al., 2004*) and InterProScan database (http://www.ebi.ac.uk/interpro/) (*Mitchell et al., 2015*). Finally, all identified genes encoding corresponding proteins were designated taking reference to Arabidopsis nomenclature (*Lin et al., 2001*).

All genome information of three species were downloaded from BRAD, including chromosome distribution, protein sequences and genomic sequences containing full coding sequences (CDS). The molecular weight (Mw) and theoretical isoelectric point (pI) of each Hsp70 protein were analyzed using the 'compute pI/Mw' tool of Expert Protein Analysis System (ExPASy: https://web.expasy.org/tools/) (*Wilkins et al., 1999*). The predicted value of the grand average of hydropathy (GRAVY) and instability index were

calculated by ExPASy. All Hsp70 protein sequences of *B. napus*, *B. rapa* and *B. oleracea* were analyzed using the Protein Subcellular Localization Prediction (WoLF PSORT: https://www.genscript.com/wolf-psort.html) (*Horton et al., 2007*) online tools in order to predict subcellular localization.

### Prediction of *cis*-acting elements in *Hsp70* gene promoters

Approximately 1,500 bp upstream sequences of the translation initiation site (ATG) were extracted from BRAD and investigated using Plant *Cis*-Acting Regulatory Element (PlantCARE: http://bioinformatics.psb.ugent.be/webtools/plantcare/html/) (*Lescot et al., 2002*), which were to determine putative *cis*-acting regulatory elements in the promoter region of *Hsp70* genes.

### Comparative phylogenetic analysis of Hsp70 proteins

Sequence alignments were performed and phylogenetic analyses were constructed to explore the evolutionary relationship of Hsp70s in *B. napus*, *B. rapa*, *B. oleracea* and *A. thaliana*. All protein sequences were performed multiple alignments in MUSCLE program of Molecular Evolutionary Genetics Analysis (MEGA 7) software (*Kumar, Stecher & Tamura, 2016*). An unrooted phylogenetic tree was constructed based on the Maximum Likelihood (ML) method, with a 1,000 bootstrap replicates and a Jones-Taylor-Thornton (JTT) model. The Interactive Tree of Life (iTOL: http://itol.embl.de/) (*Letunic & Bork, 2016*) website was used to better visualize the tree.

### Analysis of *Hsp70* gene structures and conserved domains of their encoding proteins

Using FASTA files of the coding and corresponding genomic sequences, exon-intron structures of *Hsp70* gene were determined with the Gene Structure Display Server (GSDS: http://gsds.cbi.pku.edu.cn/) (*Hu et al., 2015*). The conserved motifs of Hsp70 protein were investigated with the Multiple EM for Motif Elicitation (MEME: http://meme-suite.org/) (*Bailey et al., 2009*) tool, with parameters set as follows: minimum motif width: 6, maximum motif width: 50, and maximum number of motifs: 20; default values were used for remaining parameters. To find the conserved signature domain of Hsp70 proteins, sequence alignment of all identified Hsp70 proteins were carried out using Multiple alignment program for amino acid or nucleotide sequences (MAFFT: http://mafft.cbrc.jp/alignment/software/) (*Katoh, Rozewicki & Yamada, 2017*), and were displayed by Jalview software (*Waterhouse et al., 2009*).

### Chromosome localization and *Hsp70* gene duplication events

The position and length of *Hsp70* genes in each chromosome extracted from BRAD, then all *Hsp70* genes were mapped to specific chromosomes except some genes located on random scaffolds. MapInspect tool was used to show the location information (*Wang et al., 2019*). Duplicated *Hsp70* genes were detected using Nucleotide BLAST (BLASTn) searched against protein-coding genes and their paralogs, and complied to the following criteria: the alignable coding nucleotide sequence was covered 80% of the longer gene, as well as the identity of the alignable sequences was >80% (*Yang et al., 2008*;

*Zhou et al., 2004*). If the physical distance of two homologous genes was <50 Kilobase (kb), it was defined as tandemly duplicated genes (*Cannon et al., 2004*).

To investigate synteny relationship of closely related species, all *Hsp70* genes among *B. napus*, *B. rapa*, *B. oleracea* and *A. thaliana* were evaluated by searching "syntenic genes" in BRAD. The orthologous *Hsp70* genes located on syntenic chromosome blocks were displayed using Circos software (*Krzywinski et al., 2009*).

For estimation of selection mode for *BnHsp70*, *BrHsp70* and *BoHsp70* genes, the ratio of non-synonymous to synonymous substitutions (Ka/Ks) of all segmental gene pairs were calculated by DnaSP (*Librado & Rozas, 2009*). The value of Ka/Ks ratio >1, =1 and <1 are represented for positive selection, neutral selection and negative or stabilizing selection, respectively.

## Plant material and tissue collection

In this study, healthy seeds of *B. napus* (cv. Darmor), *B. rapa* (cv. chiifu) and *B. oleracea* (cv. Jinzaosheng) were selected for further cultivation. In the autumn of 2017, all seedlings were grown in natural environments of Wuhan University. According to the BBCH (the Biologische Bundesanstalt, Bundessortenamt and Chemical industry) scale of winter oilseed rape (*B. napus*), four tissues with the flowering phase (60–69) that included inflorescence stems, young leaves, flowers and siliques from 6-month-old plants of 10 days after pollination(DAP), were collected in the spring of 2018 (*Habekotte, 1997*; *Boettcher et al., 2016*). They were frozen in liquid nitrogen and stored at −80 °C. Three biological replicates of all samples were performed in this experiment.

## Analysis of *Hsp70* gene expression patterns in various tissues

To analyze *Hsp70* gene expression patterns of different tissues in *B. napus* and two diploid progenitors, the row RNA-seq reads were deposited in the NCBI database (accession number SRR7816633–SRR7816668) (*Li et al., 2019*). RNA extraction and RNA-seq approach were similar to a previous study (*Wang et al., 2019*). Fragments Per Kilobase of transcript per Million mapped reads (FPKM) values was calculated by RSEM (Expectation-Maximization) tool to estimate the gene expression levels (*Li & Dewey, 2011*). The specific formula is as follows: $\text{FPKM} = \frac{10^6 \text{C}}{\text{NL}/10^3}$, where C in the numerator represents the number of fragments mapped only to the gene, and N and L in the denominator respectively represent the total number of fragments mapped only to the reference genome and the number of bases in the coding region of the gene. The data were normalized in order to more intuitively compare the differences of the same gene in different samples. Heat maps were generated with Heat map Illustrator (HemI: http://hemi.biocuckoo.org/down.php/) (*Deng et al., 2014*).

## RESULTS

### Genome-wide identification of *Hsp70* genes in tetraploid *B. napus* and diploid *B. rapa* and *B. oleracea*

To systematically explore all of the *Hsp70* gene family members, 107, 39 and 33 non-redundant putative protein sequences of *B. napus*, *B. rapa* and *B. oleracea* were initially

retrieved by BLASTn program in BRAD. Additional 1, 1 and 9 proteins of three *Brassica* species were also retrieved by HMM-based search with Hsp70 domain. A total of 61, 11 and 22 sequences of *B. napus*, *B. rapa* and *B. oleracea* were discarded for lack of Hsp70-specific function domain. Eventually, 47, 29 and 20 *Hsp70* genes encoding corresponding proteins were identified in the *B. napus* and two progenitors, *B. rapa* and *B. oleracea* genomes (Table 1; Table S1). All *Hsp70* genes (eg. *BnA.Hsp70-12e*, *BrHsp70-2a* and *BoHsp70-5b*) were designated corresponding to their orthologs of *Hsp70 genes in A. thaliana* (*AtHsp70*), where the last letter in the naming was "a" meaning the highest homology with Arabidopsis, next by "b", and so on. And the capital letter A or C in the name of *B. napus* took reference to the subgenome $A_n$ or $C_n$ location. Hsp70-15s to Hsp70-17s of *B. napus* and two progenitors all classified as Hsp110s, because they had Hsp70 specific domains and their size are much larger than that of classic Hsp70s. Annotation information of all identified Hsp70 proteins were shown in Table S2. There were no orthologous *Hsp70* genes for *AtHsp70-1*, *-3*, *-18* and *-14* which were found in the genomes of *B. napus* and two diploid progenitors. Additionally, no orthologous gene for *AtHsp70-7* and *AtHsp70-6* were found in *B. rapa* genome and *B. oleracea* genome, respectively. *AtHsp70-2* had only one homolog (*BnC.Hsp70-2*) in *B. napus* genome, while *BrHsp70-2* s contained 6 members and *BoHsp70-2s* contained 4 members homologous to *AtHsp70-2*. The difference of the number of copies between *B. napus* and diploid progenitors might suggest a large gene loss event occurred in the *Hsp70* gene family during polyploidization.

The length of BnHsp70s ranged from 498 to 956 amino acids (aa), with the molecular weights varing between 54.70 kDa to 106.32 kDa (Table S3). The GRAVY value of all BnHsp70s except for BnHsp70-8s was negative, indicating that most of BnHsp70 proteins were hydrophilic and suggesting BnHsp70s possibly involved in tolerance to drought stress (*Beck et al., 2007*). Approximately 74.5% (35/47) of BnHsp70 proteins (cutoff < 40) had stable structures in a test tube and the pI value of all proteins except for except for BnC.Hsp70-6d (pI = 8.98) had low isoelectric points (pI < 7). The WoLF PSORT were used to predict the subcellular location of 47 BnHsp70 proteins; The result showed BnHsp70s were mainly localized on cytoplasm (20), followed by ER (10), then mitochondrion (seven) and chloroplast (seven), and three were predicted to located on other cellular compartments. Meanwhile, BrHsp70s and BoHsp70s of cytoplasm-localized had the largest proportion, each with 13 genes (Table S4). In *B. rapa* genome, Hsp70 proteins of ER-localized, mitochondrion-localized and chloroplast-localized were predicted that had four, four and five members, respectively (Table S4). In *B. oleracea* genome, Hsp70 proteins of ER-localized, mitochondrion-localized and chloroplast-localized were predicted that had three, two and one members, respectively (Table S4). Further, these results showed subcellular locations of BnHsp70s were basically consistent with that of the corresponding homologs in two diploid progenitors (Table S4).

## Phylogenetic analyses of Hsp70 proteins in Arabidopsis and three *Brassica* species

An unrooted phylogenetic tree was built using the alignment of a total of 114 Hsp70 amino acid sequences, which included 47 members from *B. napus*, 29 from *B. rapa*, 20

**Table 1 The *Hsp70* gene family information in *Brassica napus* (cv. Darmor-*bzh*).** All 47 *BnHsp70* genes were identified using BLASTp program (BRAD; http://Brassicadb.org/brad/) and HMM-based research (http://hmmer.org/). Details about *BnHsp70* gene information were displayed.

| Gene name | Gene ID | Chromosome | Gene position | | Intron number | Arabidopsis orthologue locus |
|---|---|---|---|---|---|---|
| | | | Start | End | | |
| *BnC.Hsp70-2* | *BnaC06g01970D* | C06 | 2740114 | 2742008 | 3 | *AT5G02500* |
| *BnA.Hsp70-4a* | *BnaA01g30490D* | A01 | 20924063 | 20926784 | 1 | *AT3G12580* |
| *BnC.Hsp70-4b* | *BnaC01g38510D* | C01 | 37542383 | 37544742 | 1 | *AT3G12580* |
| *BnA.Hsp70-4c* | *BnaA03g32320D* | A03 | 15595155 | 15597674 | 1 | *AT3G12580* |
| *BnC.Hsp70-4d* | *BnaC03g37680D* | C03 | 23044377 | 23047457 | 1 | *AT3G12580* |
| *BnC.Hsp70-5a* | *BnaC08g16850D* | C08 | 20685045 | 20687250 | 0 | *AT1G16030* |
| *BnA.Hsp70-5b* | *BnaA08g23680D* | A08 | 16771839 | 16774100 | 0 | *AT1G16030* |
| *BnC.Hsp70-5c* | *BnaC05g12240D* | C05 | 7096804 | 7098541 | 0 | *AT1G16030* |
| *BnA.Hsp70-5d* | *BnaA06g10730D* | A06 | 5644636 | 5646192 | 0 | *AT1G16030* |
| *BnC.Hsp70-6a* | *BnaC01g16200D* | C01 | 11153984 | 11157177 | 7 | *AT4G24280* |
| *BnC.Hsp70-6b* | *BnaC07g38890D* | C07 | 40064163 | 40067444 | 7 | *AT4G24280* |
| *BnC.Hsp70-6c* | *BnaC01g16210D* | C01 | 11160781 | 11163748 | 8 | *AT4G24280* |
| *BnC.Hsp70-6d* | *BnaC01g16230D* | C01 | 11165715 | 11169700 | 8 | *AT4G24280* |
| *BnA.Hsp70-7a* | *BnaA01g13780D* | A01 | 7016235 | 7022008 | 8 | *AT5G49910* |
| *BnA.Hsp70-7b* | *BnaA08g14780D* | A08 | 12392265 | 12395010 | 7 | *AT5G49910* |
| *BnC.Hsp70-7c* | *BnaC08g11440D* | C08 | 16837653 | 16840640 | 7 | *AT5G49910* |
| *BnA.Hsp70-7d* | *BnaA03g46660D* | A03 | 23955244 | 23958060 | 7 | *AT5G49910* |
| *BnA.Hsp70-8a* | *BnaAnng32550D* | Ann_random | 37160556 | 37163177 | 0 | *AT2G32120* |
| *BnC.Hsp70-8b* | *BnaC04g12620D* | C04 | 9850675 | 9852366 | 0 | *AT2G32120* |
| *BnC.Hsp70-9a* | *BnaC03g61170D* | C03 | 50146745 | 50149558 | 4 | *AT4G37910* |
| *BnC.Hsp70-9b* | *BnaC01g01110D* | C01 | 481765 | 484468 | 4 | *AT4G37910* |
| *BnA.Hsp70-9c* | *BnaA01g00190D* | A01 | 66872 | 69292 | 4 | *AT4G37910* |
| *BnA.Hsp70-9d* | *BnaA08g15870D* | A08 | 13148193 | 13150962 | 4 | *AT4G37910* |
| *BnA.Hsp70-10a* | *BnaA02g00030D* | A02 | 14834 | 18467 | 5 | *AT5G09590* |
| *BnA.Hsp70-10b* | *BnaA03g55950D* | A03_random | 489297 | 492462 | 4 | *AT5G09590* |
| *BnC.Hsp70-10c* | *BnaC03g03860D* | C03 | 1874440 | 1877474 | 4 | *AT5G09590* |
| *BnC.Hsp70-10d* | *BnaC02g00800D* | C02 | 315889 | 320718 | 8 | *AT5G09590* |
| *BnA.Hsp70-11a* | *BnaA03g14210D* | A03 | 6518376 | 6521390 | 6 | *AT5G28540* |
| *BnC.Hsp70-11b* | *BnaC03g17190D* | C03 | 8771430 | 8774421 | 6 | *AT5G28540* |
| *BnC.Hsp70-11c* | *BnaC03g20620D* | C03 | 10938331 | 10941303 | 5 | *AT5G28540* |
| *BnC.Hsp70-12a* | *BnaC06g13860D* | C06 | 16719969 | 16728133 | 5 | *AT5G42020* |
| *BnC.Hsp70-12b* | *BnaC07g48050D* | C07_random | 418733 | 421268 | 5 | *AT5G42020* |
| *BnA.Hsp70-12c* | *BnaA07g05610D* | A07 | 5907236 | 5909961 | 5 | *AT5G42020* |
| *BnA.Hsp70-12d* | *BnaA03g17100D* | A03 | 8025655 | 8028557 | 5 | *AT5G42020* |
| *BnA.Hsp70-12e* | *BnaA07g15650D* | A07 | 13485784 | 13488485 | 7 | *AT5G42020* |
| *BnA.Hsp70-13a* | *BnaA09g48560D* | A09 | 32511931 | 32514629 | 5 | *AT1G09080* |
| *BnC.Hsp70-13b* | *BnaC08g42820D* | C08 | 36848273 | 36850808 | 5 | *AT1G09080* |
| *BnA.Hsp70-15a* | *BnaA04g03290D* | A04 | 2140854 | 2144541 | 8 | *AT1G79930* |

**Table 1** (*continued*)

| Gene name | Gene ID | Chromosome | Gene position | | Intron number | Arabidopsis orthologue locus |
|-----------|---------|------------|---------------|-----|---------------|------------------------------|
| | | | **Start** | **End** | | |
| *BnA.Hsp70-15b* | *BnaA06g00870D* | A06 | 609868 | 615315 | 10 | *AT1G79930* |
| *BnC.Hsp70-15c* | *BnaC04g25190D* | C04 | 26110817 | 26117673 | 10 | *AT1G79930* |
| *BnC.Hsp70-15d* | *BnaC06g06400D* | C06 | 6884609 | 6890728 | 14 | *AT1G79930* |
| *BnC.Hsp70-16a* | *BnaCnng18070D* | Cnn_random | 16871440 | 16875094 | 8 | *AT1G11660* |
| *BnA.Hsp70-16b* | *BnaA06g07260D* | A06 | 3868981 | 3872471 | 8 | *AT1G11660* |
| *BnA.Hsp70-17a* | *BnaA03g42810D* | A03 | 21484934 | 21489471 | 13 | *AT4G16660* |
| *BnC.Hsp70-17b* | *BnaC07g50110D* | C07_random | 2420784 | 2425332 | 13 | *AT4G16660* |
| *BnC.Hsp70-17c* | *BnaC01g19960D* | C01 | 13882083 | 13886038 | 13 | *AT4G16660* |
| *BnA.Hsp70-17d* | *BnaA01g17140D* | A01 | 9007657 | 9011549 | 13 | *AT4G16660* |

from *B. oleracea* and 18 from *A. thaliana* in present study. By the topology of the ML tree and bootstrap analysis of 1,000 replicates, all Hsp70 proteins were clearly divided into six subfamilies (named subfamily A to F) in final results (Fig. 1). Subfamily A was the largest subfamily containing 32 members, while subfamily E had only 5 members which were likely to be truncated based on *A. thaliana* orthologs (*Lin et al., 2001*; *Sung, Vierling & Guy, 2001*). A total of 24 members of subfamily F were all Hsp110/SSE subfamily members which were structurally very similar to Hsp70. Subfamily B was comprised of 15 members, subfamily C consisted of 17 members and subfamily D contained 21 members. Analysis of localization prediction ascertained that Hsp70 proteins encoded by genes of subfamily A and D were located in the cytoplasm and ER. Mitochondrial and chloroplastic *Hsp70* genes clustered on subfamily C and B, respectively (Fig. 1; Table S4).

In addition, Hsp70s of *A. thaliana* distributed in all subfamilies, which also indicated all *BnHsp70* genes had orthologs in *A. thaliana* genome. The AtHsp70s for each subfamily except subfamily E matched multiple sets of orthologs from *B. napus* and two progenitors. Generally, the higher of bootstrap values within each subfamily, the more statistically reliable of the derivatively homologous pairs that were branched at the same final level. A reliable pair indicates two genes had the closest relatives which located in the end of the same branch and had high bootstrap values (>50%) in a phylogenetic tree. Within this tree, a total of 37 reliable homologous *Hsp70* gene pairs were observed, and most of them were orthologous pairs between $A_n$ and $C_n$ subgenomes of *B. napus* and their respective progenitor genomes, with 16 $A_r$-$A_n$ pairs and 11 $C_o$-$C_n$ pairs. These results supported to the gene duplication events in *B. napus* genome and indicated *Hsp70* orthologous genes of distinct subfamilies kept highly conserved in respective genome.

## Structure of *Hsp70* genes and conserved domain of Hsp70 proteins in three *Brassica* species

To better characterize the structural conservation and diversification of *BnHsp70* genes during their evolution, the exon-intron organization of individual *BnHsp70* gene in coding sequence was obtained according to subfamily membership. The number of introns varied greatly, and the arrangement of introns was complex in whole *Hsp70* gene family.

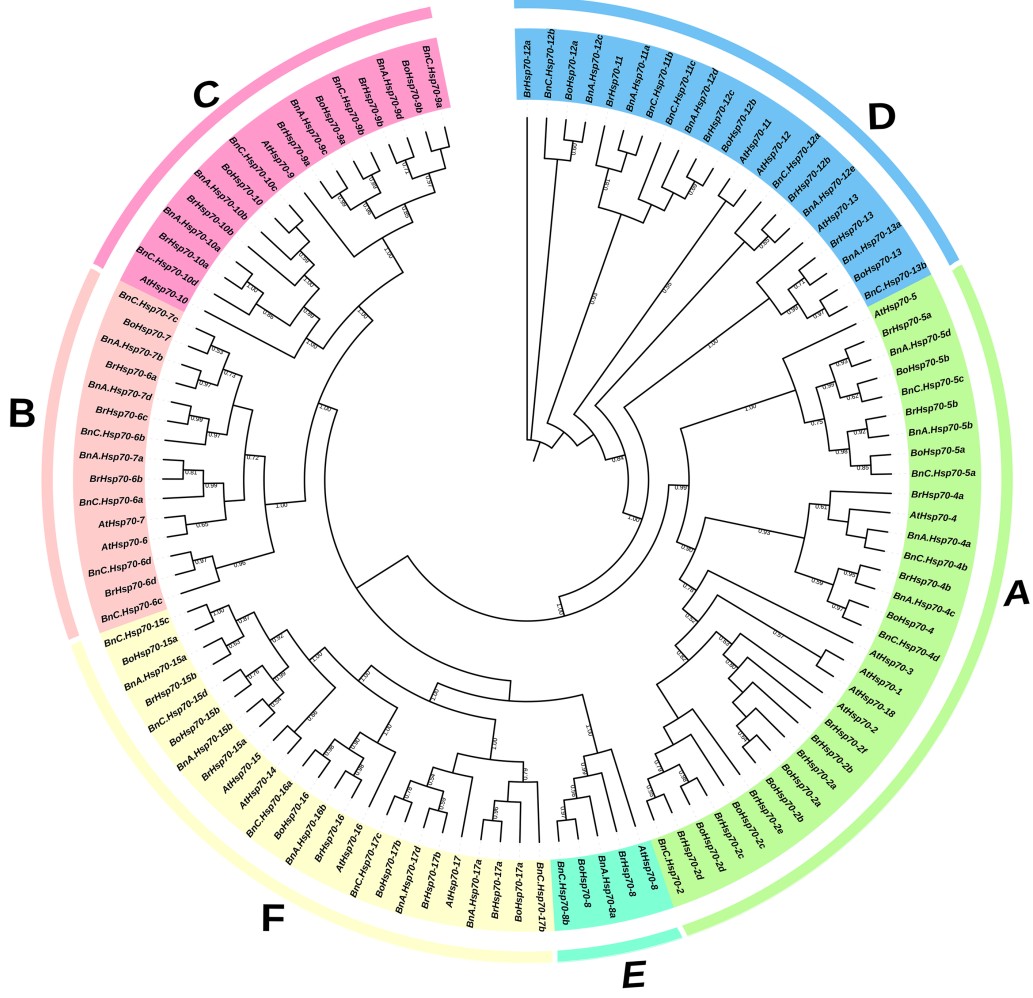

**Figure 1** **Phylogenetic analysis of the *B. napus* (cv. Darmor-*bzh*), *B. rapa* (cv. Chiifu-401-42), *B. oleracea* (var. *capitata* line 02–12) and *A. thaliana* Hsp70 proteins.** The full-length amino acid sequences of the Hsp70 proteins were aligned using MUSCLE program in MEGA 7.0. The unrooted tree was generated by the maximum likelihood (ML) method with 1,000 bootstrap replicates. All Hsp70 proteins were divided to A-F subfamilies, which were distinguished by different colors. Bootstrap values which were above 50% are indicated at the base of each subfamily.

The numbers of introns in total genes ranged from 0 to 14 (Fig. 2B). In Hsp110/SSE subfamilies, all genes had multiple introns and the highest number of intron was found in *BnC.Hsp70-15d*. The 4 truncated genes of subfamily E had no intron, whether they were members among *B. napus* or its diploid progenitors. Genes of subfamily A had zero or one intron except for *BnC.Hsp70-2*. These results suggested the gene structure within a single subfamily was highly conserved. In the course of comparison of exon-intron structure of *BnHsp70* s and two progenitor species, 25 reliable orthologous pairs were analyzed, which had high bootstrap values in a phylogenetic relationship. Approximately 40.0% (10/25) genes in *B. napus* had an identical intron number and intron phase corresponding to orthologous genes in *B. rapa* and *B. oleracea* (Figs. 2A and 2B). Other seven *BnHsp70*
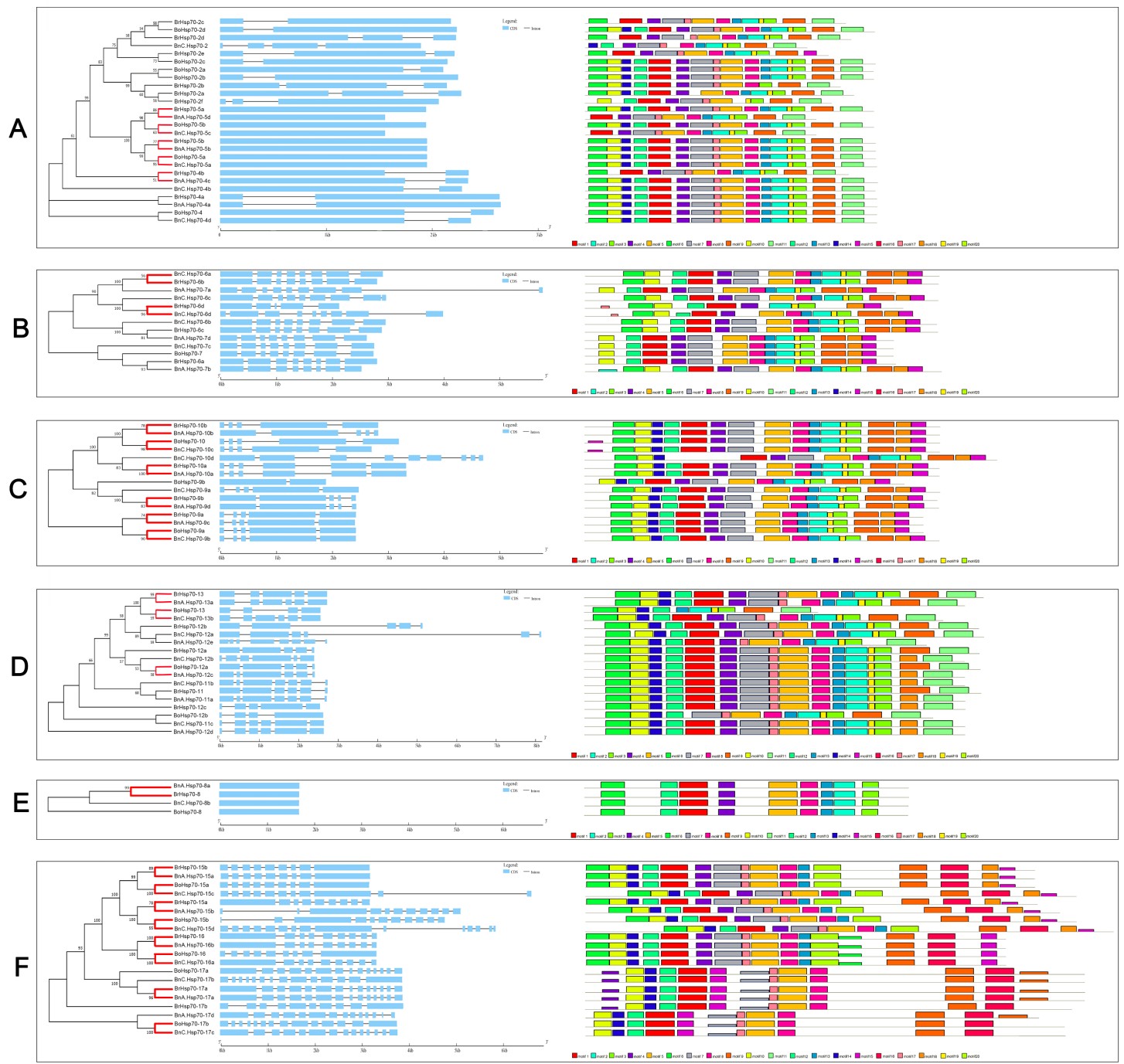

**Figure 2 Characterizations of the identified *Hsp70.* s in *B. napus*, *B. rapa* and *B. oleracea*.** The characterizations include intron/exon structure and conserved protein motif location. All Hsp70s were arranged based on similarity of amino acid sequences on each subfamily. (A) The characterizations of the *Hsp70* s in the subfamily A. (B) The characterizations of the *Hsp70* s in the subfamily B. (C) The characterizations of the *Hsp70* s in the subfamily C. (D) The characterizations of the *Hsp70* s in the subfamily D. (E) The characterizations of the *Hsp70* s in the subfamily E. (F) The characterizations of the *Hsp70* s in the subfamily F. The 25 reliable orthologous pairs between *B. napus* and two progenitors were highlighted by red branch. Blue boxes indicate exons and black lines represent introns. The gene length was estimated by horizontal axis of the bottom in the gene structure analysis (GSDS: http://gsds.cbi.pku.edu.cn/). Twenty motifs were identified through MEME analysis ( http://meme-suite.org/).

genes corresponding to their ancestral genes exhibited exon-intron loss/gain variations, and three genes changed their intron phase after allopolyploidy, while obvious differences were observed in exon lengths of 5 *BnHsp70* genes. Overall, intron numbers or phases were similar among genes with higher genetic and evolutionary similarities.

Like other identified species, the multiple protein sequence alignment of BnHsp70 family members revealed two major domains known. The highly conserved N-terminal ATPase domain contained three typical signature sequences, which were contained in approximately 400 aa (Fig. S1). Intriguingly, although the C-terminal domain was highly variable, it's exclusive and highly preserved C-terminus motif can be used to distinguish proteins of some different subfamilies. All Hsp70 proteins of cytoplasm-localized possessed signal EEVD sequence at the C-terminus. The sequences for 72.20% (13/18) ER Hsp70s and 69.20% (9/13) chloroplast Hsp70s had the conserved sequence HDEL and DVIDADFTDSK in the C-terminus, respectively (Fig. S1). However, the retention signal motif for mitochondrion Hsp70s, GDAWV and SPSQ (I/V) G, was observed in the N-terminal ATPase domain. These results suggested that the Hsp70 family was relatively conserved, while some motif sequences changed slightly during *Brassica* evolution, which possibly contributed to extended special biological function.

Using MEME, a total of 20 conserved motifs was recognized, with lengths ranging from 11 to 50 aa (Table S5; Fig. 2C). Motif 6, 4 and 5 were found in 81.3%, 89.6% and 94.8% Hsp70 proteins, whose conserved region contained conserved N-terminal domain. Three motifs created from MEME analysis results represented conserved signature sequences of Hsp70 protein specific-domain. Motif 6 contained GIDLGTT (N/Y) SCV sequences, motif 4 contained DLGGGTFDVS sequences and LVGG (S) TR (I) PKVQ sequences was included in motif 5 (Fig. 2C; Fig. S2). However, some proteins in distinct subfamily possessed preservation and expansion of specific motifs for distinguishable from those in other subfamilies. For instance, motifs 16 and 20 were uniquely found in all members of Hsp110/SSE subfamily, whereas motif 3 was absent only in this subfamily. Besides BrHsp70-2e, motif 11 was found in all members from subfamily A and D. Hsp70 members in subfamily E which were less similar to other subfamilies, contained the identical and lowest number of motifs only nine (Fig. 2C). Furthermore, 17 out of 25 orthologs in *B. napus* had a similar domain composition, which was identical to two diploid progenitors. But it seems that some BnHsp70 orthologs had truncated motifs during the allopolyploidy process, such as BnC.Hsp70-5c and BnC.Hsp70-5d lost their motif 6. These results imply that motifs containing the Hsp70-specific domains are highly conserved in all members and the type, order and number of motifs may also be used to classify different proteins for functional differences.

## Chromosomal distribution and duplication pattern analysis of *Hsp70* genes in three *Brassica* species

The chromosomal location of all *Hsp70* genes in the three *Brassica* species was investigated based on the physical position of whole genes and was shown in Fig. 3. A total of 42 *BnHsp70* s correctly mapped onto different chromosomes, excluding 5 genes located on the random scaffold of the 'Darmor-*bzh*' reference sequences. *BnHsp70* genes were clearly

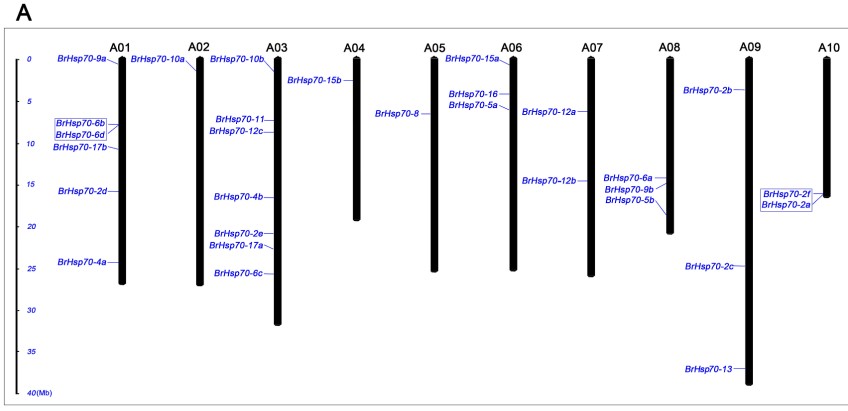

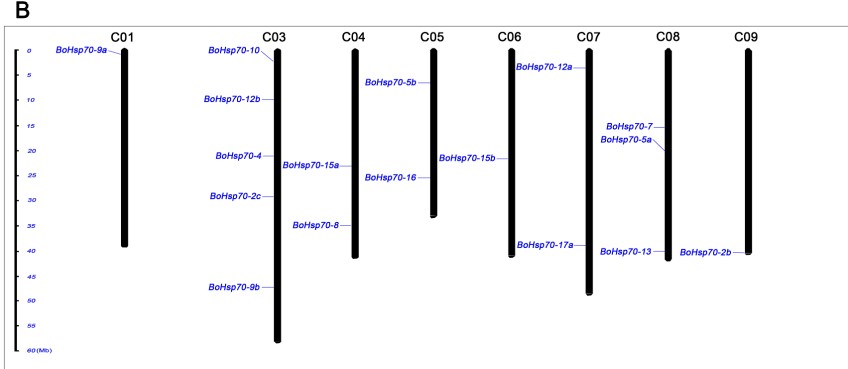

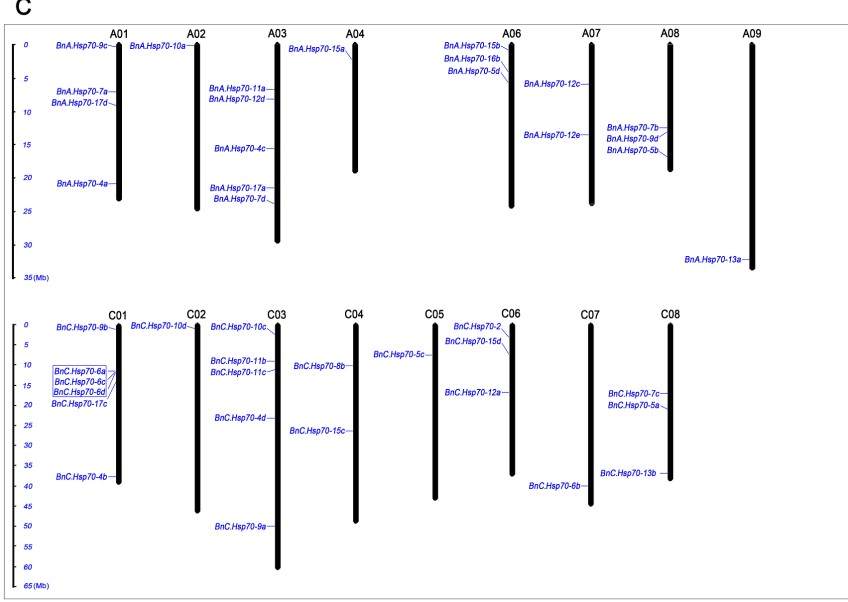

**Figure 3 Distribution of *Hsp70* gene family members on *B. napus*, *B. rapa* and *B. oleracea* chromosomes.** Distribution of *Hsp70* gene family members on *B. rapa* (A), *B. oleracea* (B) and *B. napus* (C) chromosomes. Some genes were not shown because these genes located on unmapped chromosomes. The chromosome name was indicated at the top of each bar. Tandem arrays of *Hsp70* genes were displayed within the blue frame. The scale of all chromosomes was in millions of base (Mb).

distributed across 16 of the 19, except for chromosome $A_n$ 05, $A_n$ 10, and $C_n$ 09 (Fig. 3C; Table 1). The number of *Hsp70* genes varied considerably among different chromosome. Chromosome $C_n$ 01 in *B. napus* carried the greatest gene numbers (6) and it is worth mentioning that *BnC.Hsp70-6a*, *-6c* and *-6d* in $C_n$ 01 were clustered in a sequence distance of 50kb. Moreover, 42 *BnHsp70* s had non-random distribution across 16 chromosomes, with 20 in the $A_n$ subgenome and 22 in the $C_n$ subgenome. The number of *Hsp70* genes had approximately equal distribution on the $A_n$ and $C_n$ subgenome. Furthermore, distribution of *BnHsp70* genes appeared to a consistent match with that of their orthologous genes in diploid ancestor genomes ($A_r$ genome, 29 and $C_o$ genome, 17). The distribution of 18 *BnHsp70* genes in $A_n$ subgenome was identical to orthologous gene in *B. rapa* genomes, while 11 of $C_n$ subgenome were identical to that in *B. oleracea* genome (Fig. 3). These results indicated chromosome location of *Hsp70* s might be derived from long-term gene duplication in the evolution process.

In order to better understanding *Hsp70* gene expansion and clustering, it is important to analyze chromosomal syntenic gene in *Brassica* species and *A. thaliana*. Generally, synteny analysis represented genomic fragments from different species that derived from an identical ancestor, which mainly was used to share gene annotations and reveal genomic evolution of related species (*Cheng et al., 2012*). By searching 'syntenic gene' in BRAD, a total of 63 *Hsp70* genes in three *Brassica* species showed conserved synteny with those in *A. thaliana* and were positioned in the same conserved chromosomal blocks, such as A, U, R, F, S, J and D (*Schranz, Lysak & Mitchell-Olds, 2006*). In addition, syntenic genes in three *Brassica* species were divided into three fractionated subgenomes (*Liu et al., 2014*). LF (Least-fractionated) subgenome contained 23 *Hsp70* genes, and both 20 genes were caught in MF1 (Medium-fractionated) and MF2 (Most-fractionated) subgenome (Table S6). About 65.6% (63/96) of *Hsp70* genes from three *Brassica* species was located in syntenic blocks, suggesting the expansion of *Hsp70* genes was also accompanied by gene loss. To detect the retention or loss of *Hsp70* genes after WGT and allopolyploidy events, the synteny relationship of *Hsp70* gene homologs were further visually depicted by Circos software between $A_n$ and $C_n$ subgenome of *B.napus* and two diploid progenitors, *B. rapa* and *B. oleracea*. (Fig. 4; Table S6). A total of 13 *AtHsp70* genes retained corresponding syntenic paralogs in *Brassica* species. In these genes, four *Hsp70* genes (*Hsp70-4/5/9/13*) among all three *Brassica* species were completely preserved in the same block of synteny, whose function might be enhanced adaptation of *B. napus* in an adverse environment. Interestingly, 2 of 4 *AtHsp70* genes (*AtHsp70-5/9*) were preserved as two copies among *B. rapa* and *B. oleracea* genomes and $A_n$ and $C_n$ subgenome, which were located on symmetrical subgenome (LF, MF1 or MF2). Only *AtHsp70-6* were retained as all three copies in *B. rapa* genome after triplication and maintained synteny with *BnHsp70-6s*, which might imply these genes had a unique biological function during evolution. Notably, synteny analyses implied *BnC.Hsp70-6a/6c/6d* genes might have presented tandem array, which was consistent with the chromosomal location of these genes (Table S6; Fig. 3).

Moreover, the generation and maintenance of multigene family may be significantly affected by tandem duplication and segmental duplication (*Cannon et al., 2004*). According to the descriptions (*Zhou et al., 2004*), those closely related genes with a physical sequence

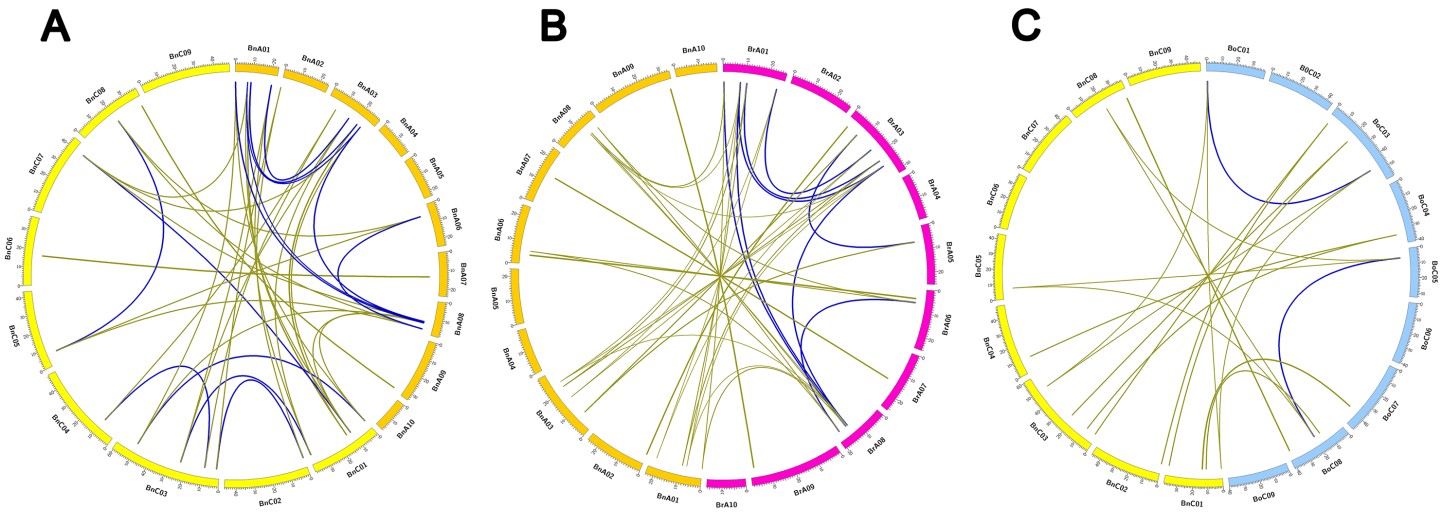

**Figure 4** **Genome-wide synteny analysis for *Hsp70* genes among *B. napus*, *B. rapa* and *B. oleracea*.** (A) Synteny analysis of *Hsp70* genes on $A_n$ and $C_n$ subgenome in *B. napus*. (B) Synteny analysis of *Hsp70* genes between $A_n$ subgenome of *B. napus* and *B. rapa*. (C) Synteny analysis of *Hsp70* genes between $C_n$ subgenome of *B. napus* and *B. oleracea*. Inside the circos, brown lines linked the syntenic orthologs and blue lines linked the syntenic paralogs.

of 50 kb were defined as tandem duplication. It was discussed that the fate of orthologous *Hsp70* gene pairs in the tandem array of *Brassica* lineages split from Arabidopsis. Only one tandem *BnHsp70* gene cluster was identified in *B. napus* genome, which was composed of *BnC.Hsp70-6a*, *BnC.Hsp70-6c* and *BnC.Hsp70-6d*. But there were two tandem duplicates in *B. rapa* genomes, *BrHsp70-2a/2f* and *BrHsp70-6b/6d* (Fig. 3A). The previous study showed that four genes (*AtHsp70-1/2* and *AtHsp70-14/15*) were considered as tandem duplicated genes out of 18 *Hsp70* genes (*Lin et al., 2001*). Two-gene tandem array (*BrHsp70-2a/2f*) in *B. rapa* had an ancient copy but have not retained in *B. napus*, which presumed those two tandem genes arose before the divergence of *A. thaliana* and *Brassica* ancestor but was lost during allopolyploidization. Another two-gene tandem array in *B. rapa*, *BrHsp70-6b/6d*, were considered as species-specific tandem duplications which may be formed by environmental selection pressures after *Brassica* speciation. They had retained their copies in *B. napus*, while the corresponding three-gene tandem array in *B. napus* located on chromosome $C_n$ 01. According to analysis, 46 *BnHsp70* genes was thought of as segmentally duplicated genes allowing the criteria described above, which were much higher than 25 and 15 duplicate genes detected in *BrHsp70*s and *BoHsp70*s, respectively. It can be concluded that segmental duplication events play a greater crucial role than tandem duplication during the expansion of *Hsp70* genes in *B. napus*.

Typically, the non-synonymous (Ka or dN) and synonymous (Ks of dS) substitution ratios were calculated to verify whether selective pressures acted on these segmental duplications. The results revealed the Ka/Ks values of all identified *Hsp70* segmental duplications were always lower than 1, indicating a purifying selection on these duplicates (Table S7). In general, the Ka/Ks values significantly lower than 0.1 suggested strong purifying selection stress and functional constraint of duplicated genes. Approximately

77.01% of *BnHsp70* segmentally duplicated genes had a Ka/Ks value less than 0.1, making the structures of these gene pairs may tend to conservation and functions tend toward similarity.

### Cis-acting elements of the *Hsp70* gene promoter in three *Brassica* species

To evaluate the potential transcriptional regulation of different *cis*-acting elements distributed in the promoters of *BnHsp70* genes, promoter sequences within 1,500 bp upstream of three *Brassica* species were investigated and *cis*-acting regulatory elements (CAREs) in these regions were explored by PlantCARE database. Mainly seventeen types of defence-related CAREs were detected in the promoters of *BnHsp70* s: hormone responsive elements (10) and environmental stress related elements (7). As showed in Table S8, the promoter regions of all *BnHsp70* members contained 1-6 hormone-related elements and 2-6 stress-related elements. ARE, essential for the anaerobic induction, was detected in 44 of 47 *BnHsp70* genes except *BnC.Hsp70-9a*, *BnC.Hsp70-15d* and *BnC.Hsp70-17c*. HSE-elements were detected in 26 *BnHsp70* promoter regions, and the highest number (5) was found on *BnA.Hsp70-4c*. Additionally, some CAREs such as MBS, TC-rich repeats and CGTCA-motif were also presented in 39, 39 and 33 promoter regions of *BnHsp70* genes, respectively (Table S8). Moreover, the promoter regions of 14 *BnHsp70* genes contained more CAREs than their orthologous genes when compared 25 of reliable orthologous *Hsp70* gene pairs. Also, four orthologous pairs (*BrHsp70-13*/*BnA.Hsp70-13a*, *BoHsp70-13*/*BnC.Hsp70-13b*, *BrHsp70-16*/*BnA.Hsp70-16b* and *BoHsp7-16*/*BnC.Hsp70-16a*) had the same type and number of CAREs. These analyses suggested that cis-elements of some *BnHsp70* genes were relatively conserved after polyploidization, and expression regulations of most *BnHsp70* genes should be more abundant in response to different stress compared with diploid progenitors.

### Expression patterns of *Hsp70* genes in different tissues of three *Brassica* species

Since *Hsp70* members participate in diverse cellular functions during normal plant growth and under abiotic stress conditions, RNA-seq data of stem, leaf, flower and silique in 47 *BnHsp70* genes were extracted (Table S9). A heat map was constructed among the examined tissues to display diverse expression levels (Fig. 5C). It is worth to mention that all *Hsp70* genes in this research except *BnC.Hsp70-6d* and *BrHsp70-6d* produced relevant gene expression data. *BnC.Hsp70-6d* and *BrHsp70-6d* lacked expression data in all samples of four tissues, illustrating that it might be a non-functional expression or have special temporal and spatial expression patterns but not be detected in this study. The heat map analysis indicated that expression of *BnHsp70* members varied greatly among tissues, holding functional diversification of the *Hsp70* genes during *B. napus* development. As showed in Fig. 5C, the majority of *BnHsp70* genes exhibited significantly tissue-specific expression patterns in all examined tissues. 6 *BnHsp70* s in leaf, 3 in stem (*BnA.Hsp70-11a/15a* and *BnC.Hsp70-15c*) and 1 in silique (*BnA.Hsp70-10b*) showed relatively high expression levels, which *BnA.Hsp70-7d* of all genes had the highest transcript abundances across four tissues. Interestingly, *BnC.Hsp70-6a/6b* and *BnA.Hsp70-7a/7b/7d* displayed

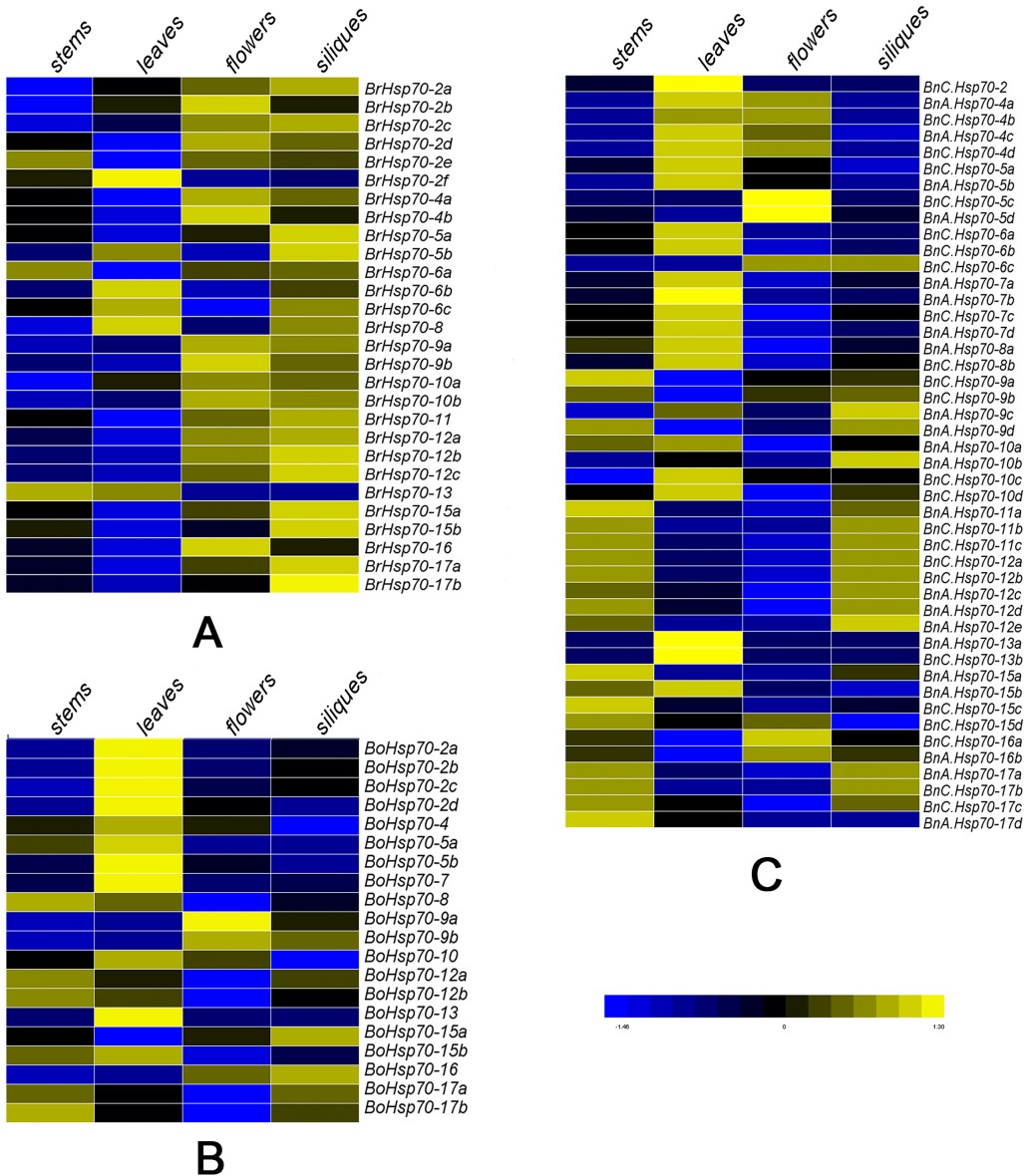

**Figure 5** **Expression patterns of *Hsp70* genes in four tissues (stem, leaf, flower and silique).** (A) Expression levels of 28 *Hsp70* genes in different tissues of *B. rapa*. (B) Expression levels of 20 *Hsp70* genes in different tissues of *B. oleracea*. (C) Expression levels of 46 *Hsp70* genes in different tissues of *B. napus*. The log-transformed values of the expression trends of *Hsp70* genes were used for hierarchical cluster analysis (original data shown in Table S9). *BnC.Hsp70-6d* and *BrHsp70-6d* were not shown because their relevant gene expression data were not detected. The color scale in the bottom represented expression levels with high transcript abundances (yellow) or low transcript abundances (blue).

high expression in leaf, suggesting that these chloroplast-localized genes may carry out related biological functions in leaf. Also, this similar higher expression pattern was also observed in different tissues. For example, *BnHsp70-4* s were highly expressed specifically

in leaf and flower, while *BnHsp70-11* s and *BnHsp70-12* s (except *BnA.Hsp70-12e*) had higher expression in stem and silique.

Furthermore, the preferential expression of *BnHsp70* genes and their homologs in related diploids was analyzed based on expression data between *B. napus*, *B. rapa* and *B. oleracea* (Fig. 5). The majority of *Hsp70* genes in the same homologous pairs displayed distinct expression patterns. For example, *BnA.Hsp70-5d* was expressed at a low level among four tissues, while *BrHsp70-5a* was a specific high expression in leaf. Likewise, the expression profiles of *BoHsp70-5b* and *BnC.Hsp70-5c* homologous pair were quite different across tissues, with a higher level in leaf and flower, respectively. Meanwhile, all seven selective *Hsp70* homologs (*Hsp70-5/9/10/13/15/16/17*) were analyzed and compared. A total of five *Hsp70* genes identified in leaf showed the bias toward $C_n$ subgenome, whereas there were no exhibited biased expression patterns distinctly in the other three tissues. These results may help contribute to functional differentiation of *Hsp70* gene, making the evolutionary success of polyploids and better coping with stresses in their natural environments.

## DISCUSSION

The allotetraploid *B. napus* were generated naturally about 7500 years ago and was generally considered to have complex relationships with its diploid progenitors, *B. rapa* and *B. oleracea* (Chalhoub et al., 2014). In our research, all 47 *BnHsp70* genes were completely identified and analyzed based on the sequencing and assembling of *Brassica* genomes, while 29 *Hsp70* genes were found in *B. rapa* genome and 20 in *B. oleracea* genome (Table 1; Table S1). The polyploid nature of *B. napus* renders expansion of the *Hsp70* gene family. Genome doubling in the form of polyploidy is followed by removal and retention of some redundant genomic material (i.e., many duplicate genes), possible variation in genomic structural characteristics and change of gene expression pattern (Adams & Wendel, 2005). These underlying mechanisms will have played to better understand ecological success and agronomic potential of polyploid species.

### Genome duplications play major roles in the expansion of the *BnHsp70* gene family

Studies have shown that members in the majority of gene family (80%) in the model plant Arabidopsis increased during evolution, which means the gene family expansion has occurred (Lespinet et al., 2002). Gene duplication events that included whole genome duplication, chromosome fragment replication and individual gene copies, are often the crucial driving force for plant gene family expansion. In our analyses, the abundance of *Hsp70* genes in *B. napus* may be the result of multiple gene duplication events. Previous studies revealed that the *Brassica* genome underwent three paleo-polyploidy events, which was the same as that of *A. thaliana*. Furthermore, *Brassica* species shared an extra WGT event since isolation from Arabidopsis (Liu et al., 2014; Chalhoub et al., 2014). *B. napus* was formed by hybridization and polyploidization between *B. rapa* and *B. oleracea* which were regarded as the two ancient polyploids (Schmidt, Acarkan & Boivin, 2001; Chalhoub et al., 2014). Compared to 18 *AtHsp70* genes, *B. napus* genome showed significantly a higher number of *Hsp70* genes (47 genes). Homology analysis suggested that each member

of 14 *AtHsp70* genes was homologous to 1–5 genes in *B. napus* genome (Table S1). For example, *AtHsp70-12* had five homologs in *B. napus*. Correspondingly, it had three and two homologs in two diploid progenitors, *B. rapa* and *B. oleracea*, respectively.

While polyploidy is a vital mechanism of gene family expansion, tandem duplication and infrequently segmental duplication are thought to commonly evaluated mechanisms for gene family copy numbers evolution and expansion (*Liang et al., 2017*). Therefore, it was assessed that roles of gene duplication events and Darwin's positive selection in the divergence of genes for understanding *Hsp70* gene family expansion (*Cannon et al., 2004*). The 42 *BnHsp70* s were correctly mapped onto 16 chromosomes, and only one tandemly duplicated gene cluster (*BnC.Hsp70-6a/6c/6d*) was found (Fig. 3C). *BnHsp70-6* s gene clustering phenomenon was also observed in synteny analysis. A total of 46 *BnHsp70* genes were established as segmentally duplicated genes in our study, which suggested segmental duplication event may be the main mechanism in the expansion of the *Hsp70* family in *B. napus*.

Altogether, whole genome triplication followed by main segmental duplication, played a major role in the expansion of *BnHsp70* gene family (*Cheng, Wu & Wang, 2014*; *Chalhoub et al., 2014*; *Liu et al., 2014*). Similar genome duplication patterns have been observed in late embryogenesis abundant (*LEA*) genes and Vicinal Oxygen Chelate (*VOC*) genes of *Brassica* species (*Liang et al., 2016*; *Liang et al., 2017*).

### *BnHsp70* gene loss of Large-scale mainly occurred in WGT

In theory, Each *Hsp70* gene member in Arabidopsis were expected to have three homologs in *B. rapa* and *B. oleracea* after WGT event, thus leading to even more homologs in *B. napus* genome (*Lysak et al., 2005*). However, only 47 *BnHsp70* members have been identified in the present study. Gene loss in large-scale had arisen on the duplicated *Hsp70* genes after genome duplication events. The synteny analysis revealed that 65.6% (63/96) of *Hsp70* genes from three *Brassica* species were located in conserved chromosomal blocks, whereas some genes were deleted. Chromosomal locations also indicated that the $A_n$ (22 genes) and $C_n$ (25genes) subgenomes of *B. napus* genome almost equaled that of two diploid species *B. rapa* (29 genes) and *B. oleracea* (17 genes) (Fig. 3; Table 1; Table S1). These results demonstrated that considerable loss of *BnHsp70* genes mainly occurred not on recent allopolyploidization from distinct diploid species, but on specific WGT which resulted in speciation and morphotype diversification of *Brassica* plants (*Town et al., 2006*). It is worth to mention that *AtHsp70-2* only had one homolog (*BnC.Hsp70-2*) in *B. napus* genome, which might be due to neutral loss of dispensable duplicates during the evolution process.

One possible explanation for gene loss could be that these genes experienced genomic reshuffling during rediploidization process after WGT. Logically, extensive chromosomal rearrangements after WGT mediated rediploidization and removed extra homologous chromosomes during long-term natural selection (*Paterson, Bowers & Chapman, 2004*; *Cheng, Wu & Wang, 2014*). The gene dosage imbalance issue might also explain gene loss after WGT. This hypothesis pointed out that some genes dose-changed after gene duplication had relatively low retention frequencies, since they potentially altered gene product concentrations (*Freeling, 2008*). Moreover, the gene balance hypothesis provided

that those genes whose products got involved in the macromolecular protein complexes, signal transduction and transcription factor complexes, are resistant to deletion, thus retained easily avoiding network imbalances caused by loss of members (*Thomas, Pedersen & Freeling, 2006*). In the long evolutionary process, this hypothesis may be supported by the preferential retention of *Hsp70* genes. Hsp70 cytoplasm-localized hold together to TPR protein which was the major substrate protein interacted with Hsp70s, revealing Hsp70 cytoplasm-localized was probably played a key role in adaptation (*Usman et al., 2017*). As important components of Hsp70s, the number (46) of cytoplasm-localized protein among three *Brassica* species is much higher than that of localized in other organelles, which may make it more preferentially retained during evolution.

### Intron gain of *BnHsp70* genes and domain loss of BnHsp70 proteins

Compared to non-orthologous gene sequences, orthologous genes tend to have more conserved intron positions (*Henricson, Forslund & Sonnhammer, 2010*). In this study, 10 out of 25 orthologs in *B. napus* that have a conserved intron number and intron phase corresponding to ancestral genes in *B. rapa* and *B. oleracea* (Figs. 2A and 2B). However, seven *BnHsp70* genes corresponding to their progenitor genes were found to have gained introns in the coding sequence, and no introns have been lost in all orthologs. This observation is suggestive of intron gain events in *Hsp70* genes during hybridization and polyploidization. Also, the rate of gain/loss in intron is higher than that of exons in view of the lower selection pressure in intron sequences (*Lin et al., 2006*). Generally, variation of the number and placement of intron is a common process that has occurred during evolution (*Roy & Gilbert, 2005*; *Jeffares, Mourier & Penny, 2006*; *Rogozin et al., 2012*). Furthermore, the factors that determine the evolutionary fate of intron count on the intron itself, the gene in which it exists and the host organism (*Jeffares, Mourier & Penny, 2006*). We suggest that intron additions of orthologs in *BnHsp70* family are a mechanism of allopolyploid adaptation, which is beneficial to conquer genomic shock generated from hybridization event that two differentiated diploid genomes reunited in a common nucleus of *B. napus* genome. Introns are essential functional components of eukaryotic genomes. Interestingly, a higher number of introns in rice can lead to a higher expression levels by providing post-transcriptional stability for mRNA (*Deshmukh, Sonah & Singh, 2016*). Thus, intron gains in 7 *BnHsp70* genes may increase the diversity of gene function to varying extents, which may have contributed to being given higher phenotypic plasticity of *B. napus* than two progenitor species. Meanwhile, it was obviously observed that the intron length of *BnA.Hsp70-4c* was truncated compared with orthologous gene *BrHsp70-4b* (Fig. 2B), which may influence their mode of expression (*Chorev & Carmel, 2012*). There are evidence that variation in the intron length appears to affect the frequency and type of alternative splicing, and longer introns are more likely to undergo alternative splicing and no splicing (*Fox-Walsh et al., 2005*; *Kim, Magen & Ast, 2007*). We think that changes in intron length may help to optimize *Hsp70* gene structure and function and facilitate the evolution of species after polyploidization. In summary, intron dynamics in *Hsp70* gene family reveal common or differing trends in *B. napus* genome evolution following polyploidy.

Previous research demonstrated that more than one-third of all domains have a marked tendency to increase/decrease in size in protein evolution statistically (*Wolf et al., 2007*). Our orthologs analyses clearly showed 17 out of 25 orthologous BnHsp70 proteins had similar motif composition, indicating conservation of domain in BnHsp70s was highly consistent with that in two diploid species and also emphasizing their close evolution relationship in three *Brassica* species. BnC.Hsp70-5c and BnA.Hsp70-5d had both lost motif 6 compared with their orthologs BoHsp70-5b and BrHsp70-5a, while BnA.Hsp70-13a had lost motif 5 compared with BrHsp70-13. Consequently, three BnHsp70 proteins lost their conserved NBD domain of fragment due to typical signature sequences included by motif 6 and motif 5. Except for the effects of erroneous annotations, we can consider domain loss of fragment represents protein evolution of BnHsp70 family in the long-term polyploidy adaptation.

### Subgenome bias of *Hsp70* genes in *B. napus* and expression of *BnHsp70* members under diverse stress

After breaking down the hybridization barrier and undergoing genomic shock, the *B. napus* genome has become a stable genome which may allow considerable subgenome interaction (*McClintock, 1984*). As one of the widespread consequences of subgenome interaction, gene conversion between two subgenomes routinely refers to transfer genetic information between genes by a unidirectional approach (*Samans, Chalhoub & Snowdon, 2017*). Using the gene conversion dataset previously published, homologous gene conversion arose in *BnC.Hsp70-6a* ($C_n$ 01) and *BnA.Hsp70-7a* ($A_n$ 01), and this result took place with the $A_n$ subgenome as a donor (*Chalhoub et al., 2014*), which were also proved by genomic distribution and synteny analysis of *BnHsp70* gene clustering. As a outcome of allopolyploidization, similar conversion tendency at the whole-genome level were described previously in *B. napus* that the significant directional bias from subgenome $A_n$ to $C_n$ was nearly 1.3 times than the other direction and the highest rearrangement frequency was also found in the homologous chromosome pair $A_n$ 01– $C_n$ 01 (*Chalhoub et al., 2014*). In allopolyploid cotton (*Gossypium hirsutum* L.), similar homologous gene conversion events occurred biasedly from the A subgenome to D subgenome of agronomically inferior (*Paterson et al., 2012*). In addition, subgenome bias was also detected for gene expression. There were a total of 5 *Hsp70* genes (*Hsp70-5/9/10/13/17*) showed the bias toward $C_n$ subgenome when the silique transcripts were analyzed by RNA-seq in our study, whereas no significant expression bias was observed in the other three tissues. This revealed a gene expression bias related to tissue-by-subgenome interactions. Allohexaploid wheat arose as hybridization and polyploidization between *Triticum turgidum* (AABB) and *Aegilops tauschii* (DD), but the previous research showed AB- and D-subgenome were globally dominant to genes participating in the development and involving in adaptation, respectively (*Li et al., 2014*). Here, gene conversion event and biased gene expression were observed in the *Hsp70* gene family of *B. napus* genome, demonstrating that subgenome bias might be prevalent influence between fused genomes by hybridization in polyploid species. This genetic bias may be contributed to polyploids survival and success, or even drive genetic diversification in polyploid species (*Otto, 2007*;

*Samans, Chalhoub & Snowdon, 2017*). As for the cause of subgenome bias or dominance, more detailed studies are expected to confirm them.

In plants, *Hsp70* genes strongly associated with various stress resistance, also play key roles in the allopolyploid *B. napus*. In *B.napus* (cv. Zhongshuang 9), the expression profile of 20-days-old siliques underlying heat response showed that there were many considerable numbers of heat-responsive genes are up-regulation or induced to expression as the heat treatment results (*Yu et al., 2014*). In particular, 18 of all 32 up-regulated *BnHsp70* genes exhibited over 10-fold increased expression, implying up-regulation or activation of *BnHsp70* genes in siliques may be important responses for the acquisition of thermotolerance during reproductive stages. In a relatively drought tolerant *B. napus* (cv. Q2), 6,018 and 5,377 differentially expressed genes (DEGs) were detected in root and leaf in response to drought stress, and all detected 12 *Hsp70* genes were up-regulated expression (*Liu et al., 2015*). Based on previous published data (*Liu et al., 2015*), combined with the potentially reliable homologous pairs shown in Fig. 2, gene expression of *Hsp70* gene family in *B. napus* under drought stress was analyzed. By comparison, we found that *BnA.Hsp70-4c* and *BnA.Hsp70-10a* exhibited up-regulated patterns in root, whereas *BnA.Hsp70-5d* and *BnA.Hsp70-8a* showed up-regulation in leaf. In summary, *Hsp70* s in allopolyploid *B. napus* are believed to be involved in the diverse stress process and provide valuable information for the further development of the adversity-resistance breeding in rapeseed.

## CONCLUSIONS

This study primarily discussed identification, phylogenetic classification, molecular evolution and gene expression analyses of the *Hsp70* gene family in *B. napus* and diploid *B. rapa* and *B. oleracea*. All of the 47 *BnHsp70*, 29 *BrHsp70* and 20 *BoHsp70* genes were identified based on the published genome sequencing results. The Hsp70 family could be classified into six subfamilies in the phylogenetic tree. By the comparison of 25 *Hsp70* gene orthologs in *B. napus* with diploid progenitors, most exon-intron distribution and conserved motifs were conserved among the same subfamilies. With large-scale gene loss during evolution, WGT and segmental duplication events contributed the most to expansion of *Hsp70* genes in *Brassica*. Expression analysis of *Hsp70* genes indicated their tissue-specific expression profiles and $C_n$ subgenome biased expression. This work facilitates future functional and evolutionary analysis of the *Hsp70* family in many polyploid species.

### Funding

This work was supported by the National Natural Science Foundation of China (31570539, 31370258). The funders had no role in study design, data collection and analysis, decision to publish, or preparation of the manuscript.

## Grant Disclosures

The following grant information was disclosed by the authors:
National Natural Science Foundation of China: 31570539, 31370258.

## Competing Interests

The authors declare there are no competing interests.

## Author Contributions

- Ziwei Liang conceived and designed the experiments, performed the experiments, analyzed the data, prepared figures and/or tables, authored or reviewed drafts of the paper.
- Mengdi Li and Zhengyi Liu contributed reagents/materials/analysis tools.
- Jianbo Wang conceived and designed the experiments, approved the final draft.

## Data Availability

Data are available at the Sequence Read Archive of National Center for Biotechnology Information Support Center under accession numbers: SRR7816633–SRR7816668.

## Supplemental Information

Supplemental information for this article can be found online at http://dx.doi.org/10.7717/peerj.7511#supplemental-information.

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
