# Peer review of "Genome-wide identification and characterization of the Hsp70 gene family in allopolyploid rapeseed (Brassica napus L.) compared with its diploid progenitors"

_PeerJ, doi:10.7717/peerj.7511_

## Round 0.1 · original submission · Major Revisions

Please take the comments from reviewers into consideration, and make a revision accordingly.

Reviewer 1 ·

Basic reporting

The professional and appropriate English language is used throughout. The figures are relevant, high quality and well described.The introduction session provided sufficient literature, while the list of references is not in the style of Peer J, the DOI for references should be added according to the guideline Authors.

Experimental design

The research question is well defined and meaningful, which contribute to exploration of the evolutionary adaptation of polyploidy, and methods described with sufficient detail and information to replicate.

Validity of the findings

The manuscript (MS)provides a genome-wide systematically analysis of Heat shock protein 70 (Hsp70) in allopolyploid rapeseed and its two diploid ancestors. In this study, 47 Hsp70 genes in B. napus genome, 29 members in B. rapa and 20 in B. oleracea were identified based on the published genome sequencing results, gene structure, chromosomal distribution and duplication pattern, cis-acting elements and expression pattern in four tissues of Hsp70 genes were conducted. It’s very interesting that this manuscript described the tissue-specific and C-subgenome biased expression of HSP70. These findings contribute to exploration of the evolutionary adaptation of polyploidy and will facilitate further application of BnHsp70 gene functions.

Additional comments

This is a carefully done study containing interesting results, I think that the research developed in this manuscript could be worth publishing after considering these points:
1) Although this is the first report of Hsp70 gene family in B. napa, there are some reports in the other crops as mentioned in the introduction. The authors should emphasize the novelty of the characters of Hsp70 gene family in B. napa compared with others.
2) As the manuscript described that Hsp70s is indispensable in plant development, as well as associate with plant stress resistance. However, authors didn’t describe the specific function of Hsp70s members involved in developmental processes and stress responses. I think it is very necessary to characterize the role of Hsp70s in development, drought resistance, heat tolerance and other abiotic stresses.
3) The organization of the abstract is a little vague and unclear. For example,I don’t think “many evidence”(line 38) appear in the abstract is appropriate, which should be replaced by the method used. Another, “all known 114 proteins” (line 37) should be indicated the genome source, which is easily misunderstood that they are derived from three species (B. napus, B. rapa and B. oleracea).
4)“Arabidopisis” should not be italic font in lines 70, 117, 128, 222, 226, 250, 376, 472, 482, 486.
5)“The orange boxes” should be “The green boxes” in Figure 2 legend.
6)The singular and plural forms of some words need to be modified. For example, in line 68 “have” should be “has”, line 69 “are” should be “is” and in line 330 “numbers” should be “number”.

·

Basic reporting

No Comments

Experimental design

No comments

Validity of the findings

No Comments

Additional comments

Authors have conducted comprehensive analysis of Hsp70 gene family in rapeseed and its related diploid species. This analysis included detail information regarding gene structure, phylogenetic analysis,chromosomal distribution, expression pattern of HSP70 gene family and have provided important information regarding its evolution. This study also provided information regarding impact of polyploidy in the expansion of HSP 70gene family. Overall this study is of high scientific value and provides new and unique information regarding important gene family.

·

Basic reporting

SPECIFIC COMMENTS:

ABSTRACT:
line 22: 47 BnHsp70 genes in B. napus. I think it is important to precise how many genes are on subgenome C or A.
line 25-26: sentence unclear: presence of orthologous (12 and 13) in which subfamiliy ?
It means also that on the 47 BnHsp70 genes only 25 are orthologous to parental genes. So, 22 are news duplicated genes in B. napus? If the case, that must be specified in the introduction. That means also that 17 B. rapa and 7 B. oleracea genes were not found in B. napus genome and lost.

INTRODUCTION:
line 52: missing space between in agriculture
At the end, the introduction needs more details :
- line 58: identification of Hsp70 genes in B. napus and parental genomes with a specific domain: which one ?
- line 71-84: What is the size of Hsp70 proteins? All the proteins have the two domains ?
- line 90-91:
duplicated in the corresponding genome : which ones ?
synteny between three genomes : which ones ? There are four genomes used : B. napus compared with A. thaliana, B. oleracea or B. rapa ?
- line 93: four tissues : indicate which one

FIGURES:
- Fig.1: in the title, put Brassica napus in italic. Precise the cultivar (referent genome).
- Table 1 and S1: Precise the cultivar (referent genome) for B. napus, B. rapa and B. oleracea
The legends of these tables could be also supplemented briefly with the web sites used to define the characteristics.
Mw is calculated for mature proteins or with the N terminal peptide signal? Same question for pI, gravy and instability index.
- Table S2: Web site must be given.
-Fig. 2:
In the legend, for me, it is not a “phylogenetic relationship” but a dendrogramm based on similarity between nucleic sequences? Why At orthologous are not indicated? Why did you put sub-families E and F together?
(A) “BnHsp70 paralogs were marked with red diamonds”. No red diamonds are present on the figure 2.
(B) The orange boxes indicate exons and black lines represent introns. TI is green boxes for exons !
Web site must be given.
On the figure, indicate in part A, the name of the sub-families (A to F). It will be clearest in comparison to Fig.1. What is the reason for the red lines (orthologs pairs?)?
- Fig S1: I have not found a legend. N-terminal ATPase domains, EEVD, HDEL, DVIDADFTDSK,GDAWV, SPSQ (I/V) G sequences and MEME sequences motifs must be indicated. What the significance of the boxes, amino acid in green, red?
- Table S3: Indicate in title Bn Bo and Br Hsp70 proteins.
The authors could indicate in addition in the table the membership of the subcategories for each MEME motif and also which ones are the HSP70 specific domains.
- Fig. 3: The legend has to be changed. There are some results inside.
Distribution of Hsp70 gene family members on B. rapa (A), B. oleracea (B) and B. napus (C) chromosomes. Some genes were not shown because located on unmapped chromosomes.
The chromosome name was indicated at the top of each bar. Tandem arrays of Hsp70 genes were displayed within the blue frame. The scale of all chromosomes was in millions of base (Mb).
- Fig.4: The title has to be changed. Synteny is the simultaneous presence on the same chromosome of two or more loci, regardless of their genetic binding. The notion of syntenia is increasingly used to describe the preservation of the order of genes between two related species. - Remove “Dark yellow and yellow colors represented An and Cn chomosomes of B. napus. Chromosomes of B. rapa and B. oleracea were represented by red and green colors, respectively. The different colors of lines indicated their syteny relationships in three species.” Give the significance of trait colors inside the circus.
- The Circos is complicated to read. May be it is possible to divide it in three circus: one with only Bn genes to see homeologs genes, one with Bn sub-genomeA and B. rapa genes, the last one with Bn sub-genome C and B. oleracea genes.
- Table S4: what is tPCK?
- Table S6: Absicsci acid to rewrite
- Fig.5 : It is not necessary to put three times the color scale. The quality of fig.5B is not good compare to the two others. In the legend, authors have to give the name of the two genes missing.

Experimental design

SPECIFIC COMMENTS ON MATERIALS AND METHODS:
This part is described with details and is very clear to understand methods used. Some minor points to improve this part :
- In the part “identification of Hsp70 gene members” (lines 100-125), authors should identify cultivar names for reference genomes (e.g. B. napus Darmor bzh …..). I assume those are the cultivars that are used for tissue collection (line 182).
- line 165: the reference of the mapping method (MApInspect tool) is missing.
- In the part “plant material and tissue collection” (lines 181-186), authors should precise the developmental stage of plant with BBCH scale.
- In part “analysis of Hsp70 gene expression…”, lines 190-191 : It is possible to indicate what are juvenile stem (top of the stem ?), young leaves and the age of siliques (10- DAP as flowers ?).
- line 194: I don’t understand the method to extracted the reads for each Hsp70 gene ? It was on annotated gene ontology ? If not, what were the sequences to specifically select each gene from RNA data set ?
- line 195: The method to obtain the relative gene expression levels must be more detailed.
- The plant material and RNA seq approach are published by the authors in “The Gene Structure and Expression Level Changes of the GH3 Gene Family in Brassica napus Relative to Its Diploid Ancestors. Ruihua Wang, Mengdi Li, Xiaoming Wu, Jianbo Wang. Genes, 2019, 10(1): 58. doi: 10.3390/genes10010058. This reference must be cited and the two parts could be reduced.
- Expression in root is not followed. It is because no HSp70 have been expressed in it?

Validity of the findings

SPECIFIC COMMENTS:

RESULTS:
genome_wide identification of Hsp70 genes
- line 208: “a total of 47 genes” : how many were identified from each approaches ? How many with a lack of Hsp70 specific domain? Does it mean that they are not functional? May be it could be important to indicate these genes in tables 1 and S1. No pseudogenes were identified?
- line 210: “The genes from BnHsp70-15s to BnHsp70-17s belonged to hsp110s”. Precise again that they are considered as a sub-class of Hsp70 (even if it is already mentioned in the introduction).
- Moreover, it seems to me important to compare the numbers of copies in B. napus genome and the parental ones. For example, no orthologous for AtHsp70-1 -3 -18 and -14 were found in parental genomes; or still, B. napus genome contains only one gene for At5g02500 but 6 and 5 copies were found respectively in B. rapa and B. oleracea genomes.
- No information is given about the similarity between genes or proteins? It will be interested to know if gene or protein sequences are similar or not to At orthologous or parental copies.
- For the names given for B. napus genes, it will be clearer to put inside information about the membership to the sub-genome A or C. For nomenclature, as recommended by Ostergaard and King (2008), gene names were designed according to the closet At sequence homology, taking into account A or C genome location and cluster memberships. For example: BnHsp70-2 becomes BnaC.Hsp70-2. What is the B. oleracea copy corresponding (a to f)?
- lines 217-222: The size comparaisons of Hsp70 proteins in terms of aa or kDa give the same informations. This part could be reduced.
- line 223: Authors has to mention that except BnHsp70-8s, all the proteins have negative gravity values.
- line 228: At the end of this section, authors have to compare with parental proteins as they did in lines 237-238.

Phylogenetic analyses
Why AtHsp70-1, -3, -18 and -14 were put on phylogenetic approach ? They are not detected in Bn, Bo and Br genomes?
- line 247: “subfamily E had only 5 members which were likely to be truncated”. What is the explanation? If it is the size, some others small proteins would be also designed as truncated.
- lines 253-256: “Integration analysis of the phylogenetic tree and subcellular localization prediction was similar to that described for other species, including A.thaliana, rice and pepper, which suggested that the Hsp70 genes have been broadly conserved in distinct taxa (Lin et al., 2001; Sarkar, Kundnani & Grover, 2013; Guo et al., 2016). ». For me, it is discussion.
- line 259: “All subfamilies except subfamily E included multiple sets of orthologs”. I think It could be nice to indicated the At orthogs for each sub-family.
- lines 262-265: Again, to clearly shown homeologs Bn gene, names have to be changed. For example, in sub-family E : BnHsp70-8b will become BnaC.Hsp7-8, BnHsp70-8a = BnaA.Hsp70-8. It would clearly show the links between Bn genes and parental ones (line 263 “12 Ar-An and 13 Co-Cn pairs”) and also if there are duplication in Bn compared to parental genomes or gene lost between parental genomes and Bn.

Structure of Hsp70 genes and conserved domain of Hsp70 proteins in three Brassica species
- line 271: and the end of the sentence, after “obtained”, put according sub-family membership.
- line 273: “The intron distribution of BnHsp70 genes was similar in the same alignment clusters”. What is the definition of a cluster ? If I look in the sub-family A the sub-group Hsp70-2, the number of introns changed. Authors have to be more precise.
- line 276: indication of the sub-category (A).
- line 298: Replace “Table S3 by Table 3, Fig.2C
- line 304: Replace “Motif 16 and 20 » by motifs 16 and 20

Chromosomal distribution and duplication pattern analysis of Hsp70 genes in three Brassica species
- line 323:” three of which formed a gene cluster” means that they are duplicated genes?
- The notion of synteny must be written in the text.
- lines 337 : reference for Schranz blocks is missing (Schranz et al., 2006).
- lines 339-341 : reference for LF, MF1 and MF2 subgenomes is missing.
- line 355: That mean “collinear regions”?
- line 377: mention of Table S5 is wrong.

Expression patterns of Hsp70 genes in different tissues of three Brassica species
- line 414 and line 416: “All genes except BnHsp70-6d were expressed in different tissues…”. “BnHsp70-6d lacked expression data in all samples of four tissues”. IT is not really the same idea. Authors have to change the sentences.
- line 416: “indicating most of them may play important part in the growth of these tissues”. Why the authors mentioned a role in growth tissues? It is because it was young tissues. If it is the case, it has to be mentioned.
- lines 426-428: “In addition, a similar higher expression pattern was also observed in different tissues, which was similar to Hsp70 paralogs in palaeo-polyploid soybean and pepper (Zhang et al., 2015; Guo et al., 2016).” For me, it is discussion and no result.

DISCUSSION
Genome duplications play major roles in the expansion of the BnHsp70 gene family
- lines 497-500: concerning importance of cytosolic Hsp70s. What abour the others species ? It is the same preferential retention process?

Intron gain of BnHsp70 genes and domain loss of BnHsp70 proteins
- line 518-519: “Also, previous researches have shown that variation of the number and placement of intron is a common process that has occurred during evolution”. Which references?
- line 526: Intron replace by intron.

Additional comments

This work concerning Hsp70 identification and comparison in three Brassica genomes is very well conducted. Hsp70 gene family is used to understand the origin and evolution in Brassica genomes, as well as the mechanism of polyploidy adaptability. The comparison with A. thaliana genes is also positive to understand evolution inside diploid and tetraploid genomes. The experimental work appears to be carefully done. The authors seems to be used to this kind of exercise with other paper in the same aim but focus on another gene family (GH3, doi: 10.3390/genes10010058).
The first part of introduction is focus on the polyploidization and consequences of this process. The second part gives the example of Brassica lineage. The interest of Hsp70 is developed in the third part. The authors cited others species and it would be interested to have more information that only gene numbers, for example, in terms of divergence of functions or expressions of A. thaliana, rice, soybean or pepper Hsp70s. The end of the introduction define clearly the rationale and specific hypotheses of the current experiments.
The material and methods, results and discussion are well organized. Concerning results, it would have been interesting to have the percentage of similarities of genes and proteins for each sub-class (for Bn, Bo, Br and At copies) to see if there is a strong conservation and divergence between related proteins and genes. For proteins functions, Hsp70 have two important domains and only the NBD domain is really highlighted. The secondary structure analysis is also missing. Concerning gene structures with various intron-exon numbers, the possibility of alternative splicing RNA is not pointed in result (but in discussion) with may be several expressed proteins for one gene…
The main critic of this paper is the choice to name the genes. It is logical to take reference to A. thaliana nomenclature. But, the relationship between Bn genes and parental genomes it is not enough put forward. The authors have to change the names according to nomenclature recommended by Ostergaard and King (2008). It will be easier to see, in all the paper, homologies or differences between conserved Bn genes and parental ones, the lost or duplicated genes (Table S4 for example), in expression patterns (example: line 436, BoHsp70-5b and BnHsp70-5c homologous pair with different nomenclatures). No pseudogenes were detected?
This work has to be accepted with minor revisions.

·

Basic reporting

In this study, authors identified the Hsp70 gene family in allopolyploid rapeseed (Brassica napus L.), and then, compared with its diploid progenitors. It was an interesting study. I think this manuscript could be published after major revision.

Overall, the manuscript is well written. However, a better discussion part will help improve the manuscript.

The manuscript style should be revised according to the journal.

The full names of abbreviation must be given at first use, please check the whole text and revise.

The manuscript should be corrected by a native English speaker, alternatively please use one of the commercial English language editing services available.

Experimental design

well design.

Validity of the findings

Conclusions are well stated.

Additional comments

In this study, authors identified the Hsp70 gene family in allopolyploid rapeseed (Brassica napus L.), and then, compared with its diploid progenitors. It was an interesting study. I think this manuscript could be published after major revision.

Overall, the manuscript is well written. However, a better discussion part will help improve the manuscript.

The manuscript style should be revised according to the journal.

The full names of abbreviation must be given at first use, please check the whole text and revise.

The manuscript should be corrected by a native English speaker, alternatively please use one of the commercial English language editing services available.

---

## Round 0.2 · Minor Revisions

Please address the remaining minor concern raised by the reviewer.

Reviewer 1 ·

Basic reporting

The manuscript has been greatly improved.

Experimental design

no more comments.

Validity of the findings

no more comments.

Additional comments

I accept most of the author's comments. The manuscript has been greatly improved and the results are very interesting. However, as we know, Hsp70s is strongly associated with plant stress resistance. in Results, I think it is very necessary to characterize the putative stress responsive functions of each Hsp70s by performaing expression analysis under diverse abiotic stresses (high temperature,high humidity and drought conditions......), or by data from public database.

---

## Round 0.3 · Minor Revisions

Although both reviewers are now happy with the submission, a final check by Dr Julin Maloof (the Section Editor for this part of the journal) has provided the following comments which must be addressed before final Acceptance:

1) Given that all three genomes analyzed are already annotated, were new genes identified? To what extent do the authors findings compare with the existing annotation? This will help others gauge the quality of the existing annotation. If no new Hsp genes were identified relative to the annotation, this paper can still be published but the title should be changed.

2) Neighbor joining is an outdated and unreliable technique. Since phylogenetic inference is part of this paper, an appropriate technique such as maximum likelihood needs to be used.

3) To what extent could genome assembly errors affect the results? For example there are a number of Hsps found in B. napus C genome that are not present in B. Oleracera. Looking at the phylogenetic tree (and assuming it is correct), in several cases these genes are mostly closely related to B. rapa and B. napus A genes...this implies that their origin predates the origin of B. napus and they would be expected to be present in B. oleracea. Examples include Bnc70-10d, 70-2, 70-15d, 70-6a, 70-6d, and maybe more.

4) The authors say that genes are named based on orthology to A.t. genes but if the phylogenetic tree is correct this does not seem to always be the case. For example, there is a clade of Brassica 70-6 and 70-7 genes that jumbled together and have A.t 70-6 as a sister to the whole clade. Shouldn't these all be 70-6 genes? Writing is not always clear for example: line 263: "High bootstrap values were showed in a large number of internal subfamilies, demonstrating statistically reliable of derivatively potential homologous pairs."

5) The paper should be reviewed by a native English speaker.

Reviewer 1 ·

Basic reporting

Ok

Experimental design

Ok

Validity of the findings

Ok

Additional comments

I accept the comments.

---

## Round 0.4 · accepted · Accept

I am pleased to inform that your manuscript is accepted for publication.